

# A novel deep learning based approach with hyperparameter selection using grey wolf optimization for leukemia classification and hematologic malignancy detection

Shams ur Rehman[1], Robertas Damaševicius[2], Hassan Al Sukhni[3], Abeer Aljohani[4], Ameer Hamza[2], Deema Mohammed Alsekait[5] and Diaa Salama AbdElminaam[6,7]

[1] Department of Computer Science, NUTECH University, Islamabad, Pakistan
[2] Centre of Real Time Computer Systems, Kaunas University of Technology, Kaunas, Lithuania
[3] Cybersecurity Department Faculty of Science and Information Technology, Jadara University, Irbid, Jordan
[4] Department of Computer Science, Applied College, Taibah University, Medina, Saudi Arabia
[5] Department of Information Technology, College of Computer and Information Sciences, Princess Nourah Bint Abdulrahman University, Riyadh, Saudi Arabia
[6] Faculty of Computers and Artificial Inellgence, Benha University, Benha, Egypt
[7] Jadara Research Center, Jadara University, Irbid, Jordan

Corresponding authors
Ameer Hamza,
ameer.hamza@ktu.edu
Deema Mohammed Alsekait,
Dmalsekait@pnu.edu.sa

## ABSTRACT

Traditional diagnostic methods of leukemia, a blood cancer disease, are based on visual assessment of white cells in microscopic peripheral blood smears, and as a result, they are arbitrary, laborious, and susceptible to errors. This study proposes a new automated deep learning-based framework for accurately classifying leukemia cancer. A novel lightweight algorithm based on the hyperbolic sin function has been designed for contrast enhancement. In the next step, we proposed a customized convolutional neural network (CNN) model based on a parallel inverted dual self-attention network (PIDSAN4), and a tiny16 Vision Transformer (ViT) has been employed. The hyperparameters were tuned using the grey wolf optimization and then used to train the models. The experiment is carried out on a publicly available leukemia microscopic images dataset, and the proposed model achieved 0.913 accuracy, 0.892 sensitivity, 0.925 specificity, 0.883 precision, 0.894 F-measure, and 0.901 G-mean. The results were compared with state-of-the-art pre-trained models, showing that the proposed model improved accuracy.

# INTRODUCTION

Leukemia is a malignant disorder of the blood and bone marrow considered by the unregulated proliferation of abnormal white blood cells (leukemic cells) (*Talaat & Gamel, 2024*). The normal function of blood cells is often affected depending on the number of leukaemic cells. When white blood cells are in excess, the body's immune system is compromised; when red blood cells are in abundance, the transport of oxygen to the body

is impaired; and an increased number of platelets can result in bleeding disorders (*Habchi, Bouddou & Aimer, 2024*). Leukemia is generally classified into four types: acute lymphoblastic leukemia (ALL), acute myeloid leukemia (AML), chronic lymphocytic leukemia (CLL), and chronic myeloid leukemia (CML). Although all four types of leukemia affect children and adults. The clinical presentation and pathology of the disease differ among the four types (*Al-Bashir, Khnouf & Bany Issa, 2024*). The cause of leukemia appears to be a combination of genetic predisposition and environmental factors in the child, adolescent, or adult who is diagnosed. Symptoms of leukemia differ depending on the kind of leukemia, but common general symptoms include: fatigue, recurrent fever, recurrent infections, unexplained weight loss, bruising, bleeding, bone pain, and swollen lymph nodes. Early detection and diagnosis are important to survival; however, most individuals have non-specific symptoms which leads to late diagnosis of the disease. After conducting the non-specific screening, the World Health Organization (WHO) and GLOBOCAN findings in 2023 show approximately 474,519 new cases of leukemia were noticed with an estimated death toll of 311,594 (*Xu et al., 2024*) globally regarding leukemia in a single year (*Ramesh & Thouti, 2024*; *Hassan, Saber & Elbedwehy, 2024*). The estimate of cases is likely to increase in 2024 due to increased odds of exposure to environmental carcinogens, and an aging population which will likely affect detection and increase treatment options, is important to enable fast detection and proper treatments (*Mafi et al., 2023*).

Leukemia has been identified through conventional methods of patient physical examination, complete blood counts (CBC), bone marrow biopsy, and microscopic examination (*Tripathi & Chuda, 2025*). The microscopic examination of peripheral blood smears and bone marrow aspirates has been the gold standard for identifying individuals' morphological features of leukemic cells. Pathologists examine the cell shape, size, nucleus-to-cytoplasm ratios, and granularity to identify leukemia subtype. Consequently, for more refined diagnosis, immunopheno typing and cytogenetic testing have identified certain surface markers and specific chromosomal translocations (*Shah et al., 2021*). However, these methods, along with immunophenotyping testing methods, are often time-consuming and can be influenced by inter-observer variability. Classifying leukemic subtypes can be a difficult process (*Jaime-Pérez et al., 2019*). Dxigital images taken by light microscopy of blood smears and bone marrow have provided access to automated procedures in addition to human examination. The automated procedures allow the use of computational tools and methods for subsequent analysis allowing for improved efficiency, reproducibility and accuracy when diagnosing cases of leukemia (*Oybek Kizi, Theodore Armand & Kim, 2025*).

Artificial intelligence (AI) has significantly influenced leukemia diagnostics in the past few years, especially in the classification of microscopic images (*Achir et al., 2024*). In leukemia diagnosis, many machine-learning (ML) approaches including support vector machines (SVM) (*Vogado et al., 2018*), k-nearest neighbors (KNN) (*Daqqa, Maghari & Al Sarraj, 2017*), and random forests (*Gupta et al., 2024*) have been used for classification of leukemic cells using hand-crafted features such as texture, shape descriptors, and color

histograms. These hand-crafted features are extracted either manually or using algorithms, rule-based, which are then fed to classifiers to differentiate between healthy and malignant cells (*Nair & Subaji, 2023*). While ML-based approaches can show promising results, their functions ultimately rely on feature engineering expertise with limited understanding of the context of the images and lack of generalization across datasets. ML techniques might not necessarily capture complicated patterns or changes in cell morphology that are necessary for accurate classification (*Ochoa-Montiel et al., 2020*; *Attallah, 2024*).

The limitations of ML have prompted the rapid transition to deep learning (DL) with convolutional neural networks (CNNs) and Vision Transformer (ViT) networks being widely adopted for DL tasks. CNNs are capable of identifying hierarchical features of input images, processing raw images as pixel values to learn features around them, CNNs have the ability to learn low-level details and high level of abstraction from the images (*Jiang et al., 2021*). This means there is less reliance on hand crafted features, providing the network with the ability to learn adapted patterns of leukemia. ViT methods (*Ahmed et al., 2025*), with self-attention mechanisms, have shown tremendous success very recently as they are great at modeling long-range dependencies and they offer a holistic view of the image to be inputted. Unlike CNN methods, ViT methods can attend to every part of the image at the same time making them fertile when understanding spatial relations of hand cell images. The emergence of CNNs and ViTs has greatly contributed to the accurate, sensitive and robust automation methods for leukemia diagnostic systems which prompts clinicians to work and make decisions more effectively (*Rezayi et al., 2021*; *Nasif, Othman & Sani, 2021*; *Othman et al., 2020*).

Recent studies have implemented several methods to classify leukemia disease using deep learning. *Prellberg & Kramer (2019)* presented a technique for recognizing leukemia using customized CNN. The authors utilized CNN-based approaches of ResNeXt with squeeze and excitation modules, which achieved 88% accuracy. This work's primary limitation was utilizing various raw image datasets and ensuring the process was transparent and repeatable. *Kumar et al. (2018)* suggested a deep learning method for classifying leukemia cancer. The authors employed a preprocessing method for reducing the noise and blurring effect. The preprocessed images were further used for the feature extraction. The authors extracted the manual features, which included color, geometry, textural, and statistical aspects. They employed naive Bayes and k-nearest neighbor and neural network. They achieved the highest accuracy on neural networks, which was 92.8%. *Setiawan et al. (2018)* suggested an automatic framework for AML subtypes M4, M5, and M7 cells. The authors used the k-means method for the segment of the cells. After that, six statistical characteristics were obtained and used in the multi-class SVM classifier's training, and the authors achieved 87% accuracy for the segmentation and 92.9% for the classification accuracy. The limitation was the less amount of data for the training process. *Laosai & Chamnongthai (2014)* designed a contour signature and k-means-based AML classification scheme for segment cells. They employed an SVM classifier and achieved up to 92% experiment accuracy. *Mourya et al. (2018)* suggested a method for classifying leukemia cancer using deep learning. The authors employed a CNN hybrid model from the

experiment; they achieved 94% accuracy. The drawback of this method was implementing GAN to produce new samples for the training, but the image quality was not good enough for the better learning of a deep learning model. *Abunadi & Senan (2022)* suggested an automatic technique for identifying leukemia cancer using hybrid models. The authors utilized CNN-based models of Resnet-18 and Google-Net. For this experiment, the authors gained 90% of the highest accuracy. The drawback of this work was the lack of sufficient data to train CNN models, which necessitates a big dataset to prevent over-fitting.

*Yadav (2021)* suggested feature fusion-based deep learning models that utilized two different convolutional neural networks (CNN); SqueezeNet and ResNet-50 to increase the classification accuracy of leukemia cells. The authors fused features from both models rather than concatenating features as in the original way feature fusion models were constructed, resulting in more complementary features representations. The dataset contained 12,500 pictures of four classes of white blood cells (eosinophils, lymphocytes, monocytes and neutrophils), where they scaled them to 150 × 150 pixels. They designed their dataset with strong design characteristics, of which they employed 5-fold cross-validation. The proposed model achieved a classification accuracy of 99.3% with precision, recall, and F1-scores of greater than or equal to 98% returned at each fold. The ROC analysis returned very high sensitivities such as the monocyte class was AUC-100 and precision for all classes. The study only noted a little confusion when classifying morphologically similar cells–particularly neutrophils and eosinophils–during the early epochs. The study does not report on the evaluation of the model in real-time or the generalization of the model as a tool with clinical applicability. *Aadiwal, Sharma & Yadav (2024)* suggested the technique of late fusion and a hybrid method employing deep learning to detect blood cancer, while incorporating both the AlexNet and VGG16 models. The hybrid model was intended to take advantage of the spatial learning elements of AlexNet and the deep-feature extraction of VGG16. They used a fairly modest-sized dataset of labeled blood cell images that had been pre-processed and augmented, but failed to provide the compositional elements. The hybrid model was trained up to 100 epochs, using early stopping, and included the Adam optimizer, to generate a converged model with no overfitting. The authors reported an accuracy of 98%, and consistently high F1-score, precision, and recall for the four types of blood cells, which led to the authors concluding that the hybrid deep learning model had produced a robust classification performance. However, the authors did not benchmark their hybrid model against modern architectures such as Vision Transformers, EfficientNet ASC, nor did they show real-time inference in the clinical context. *Yadav et al. (2023)* presented a new deep learning architecture referred to as 3SNet for the morphological diagnostics of hematologic malignancies. The new 3SNet model combined multiscale feature fusions and processed three types of input images (grayscale, local binary patterns (LBP) and histogram of oriented gradients (HOG)) that incorporates texture and shape alterations in leukemic cells. This was particularly developed to meet some of the challenges of morphologically and functionally similar cell types as well as class imbalance. The model used the AML-Cytomorphology-LMU dataset for training and testing, which contains labeled

images of several leukemia sub-types. As judged by the mean average precision (MAP), the model exhibited a MAP of 93.83% for well represented classes but a MAP of 84% for under represented classes. The model exhibited area under the receiver operating characteristic curve (ROC AUC) scores above 98% in most classes. These results provide evidence that the model has the ability to differentiate leukemia cells, even with limited data. The limitation of the method was the moderately high computational cost due to processing triple input, as well as the lack of testing on actual real-world clinical data to establish its generalization. *Tanwar et al. (2025)* suggested a hybrid deep learning model, ResViT, in which ResNet-50 was combined with a two-views ViT for effective leukemia cell diagnosis based primarily on 20,000 imagery findings associated with the different stages of leukemia and secondly with an 18,236 single-cell image dataset comprising of 15 morphological classes. The ResViT model utilized convolutional and transformer streams for extracting both local level spatial features and global contextual dependencies. The ResViT model performed above 99% accuracy on both datasets outperforming stand-alone CNNs or ViTs. The use of both local and global attention allowed ResViT to appropriately avoid challenges associated with subtle morphological differences or contaminated and noisy data. However, limitations to ResViT included inability for deployment in a lower-resourced cognitive clinical setting, due to an overly complex architecture and that there was no output inference time or deployment capability reported to enable rapid fire head-and-face schema diagnosis. *Kadry et al. (2022)* proposed a CNN-assisted segmentation approach for leukocyte extraction from RGB-scaled blood smear images using models such as SegNet, U-Net, and VGG-UNet. The study used the LISC dataset and demonstrated that the VGG-UNet model outperformed others with a Jaccard Index of 91.51%, Dice coefficient of 94.41%, and accuracy of 97.73%. The robustness of the approach was further validated using additional datasets like BCCD and ALL-IDB2. *Maqsood et al. (2025)* introduced Csec-net, a deep feature fusion and entropy-controlled firefly optimization-based feature selection framework for leukemia classification. Their method involved preprocessing, transfer learning with five CNN architectures, feature fusion *via* convolutional sparse decomposition, and selection using a firefly algorithm. Classification was performed using a multi-class SVM, achieving high accuracies across four datasets upto 99.64% on ALLID_B1 demonstrating the efficacy of the proposed pipeline.

Challenges still remain in the classification of leukemia in images viewed with a microscope such as, such as classifying leukemia cells from images is the bizarre morphology and multi-nucli features of malignant cells. For example, unlike normal blood cells, leukemia cells are associated with having larger nuclei that can appear asymmetric and extremely large nucleoli that vary in size, shape, and texture depending on the patient. The variations in complexity of the structures are difficult for traditional CNNs to recognize because they rely on narrow convolutional filters that may not be able to discern long-term dependencies, much less subtle common global patterns. Therefore, in this research, we propose an automatic lightweight deep learning framework based on a novel CNN model named PIDSAN4 and modified ViT for classifying leukemia disease using microscopic images.

The main contributions are as follows:

- A novel preprocessing pipeline that includes a hyperbolic sin based contrast enhancement that boosted the visibility of microscopic blood smears to improve the classification performance.
- A custom parallel inverted dual self-attention network (PIDSAN4). It uses lightweight inverted residual blocks combined with self-attention mechanisms to represent the irregular morphology of leukemic cells. The parallel combination of customized CNN and attention, has not been explored for leukemia classification previously.
- The hyperparameters of the proposed models are optimized using the grey wolf optimization method. The optimal hyperparameters are employed for the training of both proposed models.

The manuscript is organized as 'Proposed Methodology' describes the methodology based on dataset collection and preprocessing, proposed model designing, hyperparameters tuning, and training. 'Experimental Results' discussed and compared the results of the proposed framework. 'Comparison with Sota Methods' concluded with the discussion of the achieved results.

## PROPOSED METHODOLOGY

### Dataset collection and preprocessing

In this research, we employed the leukemia dataset for experimental purposes. The selected dataset is publically available at https://www.kaggle.com/datasets/andrewmvd/leukemia-classification. The dataset is collected from 118 patients, and it has two classes, including normal cells and leukemia blast. The normal class contains 5,531 microscopic images, and the leukemia blast consists of 5,530 microscopic images. Each sample in the dataset has the dimension of $224 \times 224 \times 3$. The information on images is not clear enough to perform accurate classification. Therefore, we proposed a new technique to improve the contrast of input images. The proposed method starts by applying a hyperbolic sin function to provide a simple modification. The function is mathematically formulated as:

$$\int \sinh = \frac{e^{f(x,y)} - e^{-f(x,y)}}{2} \tag{1}$$

where $f(x, y)$ represents the low contrast input image, and $\int \sinh$ denotes the output of the hyperbolic sine function. After that, the power law function is performed on the resultant image. The power law is defined as:

$$\partial_p = \rho \cdot \left( \int \sinh \right)^{\alpha} \tag{2}$$

where $\partial_p$ denotes the resultant image of the power law, $\rho$ represents the controlling parameter of brightness, and $\alpha$ controls the enhancement in the contrast. Following that,

the hazing layer is removed by employing the dehazing function. The mathematical equation of dehazing is:

$$\phi_h = \frac{\partial_p - \omega_A}{\varepsilon_t} + \omega_A \tag{3}$$

where $\phi_h$ denotes the output image of dehazing, $\omega_A$ represents the estimation of atmospheric light, and $\varepsilon_t$ is the estimation of the transmission map. In the last phase, the resultant image is passed through the contrast stretching function to remap the pixels into their dynamic range. The function is described as:

$$F_{\text{output}} = \frac{\phi_h - \min(\phi_h)}{\max(\phi_h) - \min(\phi_h)} \tag{4}$$

where $\max(\phi_h)$ and $\min(\phi_h)$ represent the maximum and minimum intensity values, and $F_{\text{output}}$ is the final output of the proposed algorithm, which is visually presented in Fig. 1.

## Proposed parallel inverted dual self-attention network (PIDSAN4)

CNNs are the baseline architecture for numerous computer vision tasks, including image classification, object detection, and segmentation. CNNs are structured to automatically and adaptively learn different spatial hierarchies of features from the input; the fundamental property in the standard architecture of a CNN is the convolutions; the convolutions are applied to the input *via* multiple filters which capture the local patterns in the input like edges, corners, textures, *etc*. Activation functions such as rectified linear unit (ReLU) are applied after the convolutions to introduce non-linearity to the model. The convolutional layers are typically followed by pooling layers which help to downsample the spatial resolution of the input in order to reduce computational costs, as well as fully connected layers which help the model to reason about high level representations for classification. Although CNNs can extract features locally, CNNs are limited when it comes to modelling the long-range dependencies and global contextual features. To address this, hybrid models that integrate CNNs with self-attention techniques. Hybrid models seek to capitalize on the local feature extraction capabilities of CNNs, alongside the global modelling capabilities of self-attention.

Pre-trained models can be an efficient and time-saving solution to many problems. Still, they also come with a certain set of limitations that disqualify their use in some applications. Such as, they have high numbers of parameters, translating into excessive computational resource requirements and memory usage that make them harder for devices with limited hardware capabilities, unable to effectively learn long-term dependencies in data, as they are usually intended to solve generic tasks like object recognition, and fixed architectures can limit the adaptability to learn unique features. Therefore, it is necessary to optimally design CNNs to realize these desired features to solve specific tasks. In the work, we designed a lightweight hybrid model based on parallel inverted residual blocks integrated with self-attention mechanism named with PIDSAN4. The purpose behind designing the PDSAN4 is to learn the irregular shape and large nuclei,

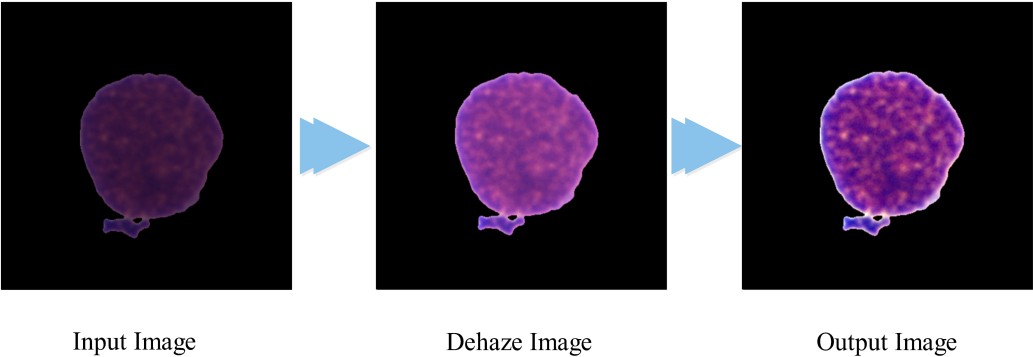

| Input Image | Dehaze Image | Output Image |

**Figure 1 Visual representation of the contrast improvement algorithm.**

as in leukemia cells; the nucleoli are prominent compared to normal cells. The proposed model comprises 107 layers and a total of 5.7 million parameters.

**Input and initial layers:** The network starts with the input size of $224 \times 224 \times 3$. Initially, two convolutional layers are attached with the configurations of $3 \times 3$ kernel size, $2 \times 2$, $1 \times 1$ stride, and 32, 64 number of channels respectively.

**Parallel inverted bottleneck block:** After initial layers, inverted residual block has been employed with parallel branches. Each block contains one expansion convolutional configured with $1 \times 1$ filter size, 64 depth, and $1 \times 1$ stride, two batch normalization layers, swish activation, and one depth-wise convolutional operation with $3 \times 3$ filter size, 64 group channels, and $1 \times 1$ stride settings, one projection convolutional operation with $1 \times 1$ filter size, 64 depth, and $1 \times 1$ stride. A skip connection is established among the expansion to projection convolutional operations. The mathematical formulation of this block is:

$$\phi_{\exp} = \varphi(T \times \omega_{\exp}), \quad \omega_{\exp} \in \mathbb{R}^{1 \times 1 \times D \times D_{\exp}} \tag{5}$$
$$\varphi(k) = k \cdot \sigma(k) \tag{6}$$
$$\phi_d = \varphi(\phi_{\exp} \times \omega_d), \quad \omega_d \in \mathbb{R}^{3 \times 3 \times D_{\exp} \times 1} \tag{7}$$
$$\phi_p = \phi_d \times \omega_p, \quad \omega_p \in \mathbb{R}^{1 \times 1 \times D_{\exp} \times D_o} \tag{8}$$
$$\phi_{\text{skip}} = T + \phi_p \tag{9}$$

where $\phi_{\exp}$ represents the expansion operation, $\varphi$ is swish activation, $\phi_d$, $\phi_p$ represent the depthwise and projection operations, and $\phi_{\text{skip}}$ is the skip connection. **Intermediate Layers:** After each parallel inverted bottleneck, max pooling activation is employed to downsample the spatial dimension with swish activation function.

$$\phi_{\boxplus} = \varphi(\max(\phi_{\text{skip}})) \tag{10}$$

**Patch embedding and dual attention layers:** After the fourth intermediate layers, the patch embedding layer is employed to convert 2D tensor into 1D sequence of patches. The

number of patches is 16. After that, the patches are passed to the dual attention layers. The first self-attention layer is applied along the spatial features and second attention is employed on channel attention. The mathematical formulation is:

$$\phi_{\boxplus} = \{p_1, p_2, p_3, \ldots, p_{16}\} \tag{11}$$

$$p_i = \text{Flatten}(P_i) \tag{12}$$

$$p_i = \omega \cdot p_i + b \tag{13}$$

$$\phi_E = [p_1 @ p_2 @ \ldots @ p_{16}] \in \mathbb{R}^{N \times D} \tag{14}$$

$$\phi_{\text{att}}^s = \text{Softmax}\left(\frac{QK^T}{\sqrt{d_k}}\right)V \tag{15}$$

$$\phi_{\text{att}}^{\top} = \text{Softmax}\left(\frac{Q_D K_D^T}{\sqrt{d_k}}\right)V_D \tag{16}$$

where $\phi_{\boxplus}$ represents the max pooling operation, $\phi_E$ is the patch embedding and $\phi_{\text{att}}^s, \phi_{\text{att}}^{\top}$ denote the spatial attention and channel attention operations. In the final step, the dual attention is presented as:

$$\phi_{\text{dual}} = \phi_E + \alpha \cdot \phi_{\text{att}}^s + \beta \cdot \phi_{\text{att}}^{\top} \tag{17}$$

where $\alpha$ and $\beta$ are learnable parameters with value of 0.5 for equal weights.

**Final layers:** After the dual attention mechanism, a 1-D global average pooling layer is utilized, followed by a new fully connected layer, a new softmax layer, and a classification layer at the end for class prediction. The loss function of the proposed model is categorical cross-entropy to minimize the training loss, defined as:

$$\mathscr{L} = -\sum_{i=1}^{N} y_i \log(\hat{y}_i) \tag{18}$$

$$\psi_{\text{Loss}} = -\sum_{\alpha=1}^{N} \sum_{\beta=1}^{C} O_{\alpha\beta} \log(P_{\alpha\beta}) \tag{19}$$

where $N$ and $C$ represent the number of samples and classes, respectively. $O_{\alpha\beta}$ denotes the $\alpha$-th sample belonging to the $\beta$-th class, and $P_{\alpha\beta}$ is the predicted probability output. Figure 2 presents the proposed model architecture.

## Modified vision transformer

ViT, a deep learning architecture, draw inspiration from the Transformer model, initially designed for text data within natural language processing (NLP) settings, to tackle image processing tasks. ViT is selected to take advantage of its long-range dependency processing features and flexibility in handling different features, making it suitable for identifying complex and subtle features of leukemia and normal cells. The tiny16 variant is employed in this work due to its lightweight complexity and computational cost.

The proposed ViT is briefly described as follows: Suppose $T = \{(A_i, b_i)\}_{i=1}^s$ is a set of $s$ images, where $A_i$ denotes a sample image and $b_i$ denotes the corresponding class label of that image, $b_i \in \{1, 2, \ldots, n\}$, where $n$ represents the number of classes in set $T$

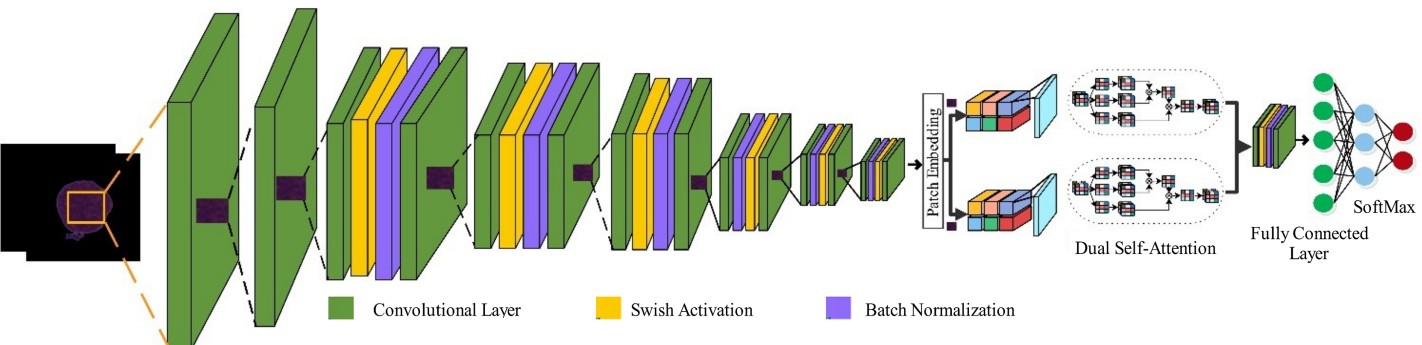

**Figure 2** High-level architecture of the proposed PIDSAN4 model.

(*Bazi et al., 2021*). The ViT model aims to learn a mapping from patterns of image patches to their semantic labels. The architecture of ViT is inspired by the vanilla Transformer, which follows the encoder-decoder architecture, capable of processing sequential data independently of recurrent networks (*Vaswani, 2017*; *Holliday, Sani & Willett, 2018*).

The self-attention mechanism is a key strength of the transformer model, as it acquires relationships between elements in long-range sequences. ViT extends transformers for image classification. The attention mechanism enables ViT to process different regions of an image and concatenate information obtained from the entire image sequence. The generic architecture of the ViT model comprises multiple layers, including an embedding layer, an encoder, and a classification layer.

Initially, a sample image $A_i$ is split into patches, each containing non-overlapping information of the image. Each patch is considered a token by the transformer. For an image $A_i$ of size $(d \times t \times w)$, where $d$ is the depth of the channels, $t$ is the height, and $w$ is the width, patches of dimension $(d \times p \times p)$ are extracted. A sequence of image patches is formulated as $(a_1, a_2, \ldots, a_m)$ of length $m$, where $m = \frac{t \cdot w}{p^2}$. Commonly, a patch size of $16 \times 16$ or $32 \times 32$ is used. Smaller patch sizes increase the sequence length and *vice versa*.

The created patches are fed into the encoder after being linearly projected into a vector space of dimension $k$ through a learned embedding matrix $F$. These embeddings are concatenated with a learnable classification token $w_{\text{class}}$, whose purpose is to perform classification. The transformer visualizes the embedded patches as a set of unordered patches. To preserve spatial information, patch positions are encoded and added to the representation of the patches. The sequence of embedded patches with positional encoding is given by:

$$x_0 = [w_{\text{class}}; a_1 F; a_2 F; \ldots; a_m F] + F_{\text{pos}}, \quad F \in \mathbb{R}^{(p^2 d) \times k}, \ F_{\text{pos}} \in \mathbb{R}^{(m+1) \times k}. \tag{20}$$

Here a 1-D positional encoding scheme is utilized to preserve patch positions (*Dosovitskiy et al., 2020*). The sequence $x_0$ is fed into the transformer encoder. The encoder comprises identical layers $I$, where each layer has two sub-components: a multi-head self-attention (MSA) block and a fully connected feed-forward dense block (MLP). The MLP block consists of two dense layers with a GeLU activation function. Both

sub-components use skip connections followed by a normalization layer (LN). The operations are defined as:

$$x'_f = \mathrm{MSA}(\mathrm{LN}(x_{f-1})) + x_{f-1}, \quad f = 1, \ldots, I \tag{21}$$

$$x_f = \mathrm{MLP}(\mathrm{LN}(x'_f)) + x'_f, \quad f = 1, \ldots, I. \tag{22}$$

In the encoder, the first element from the final layer $x_I^o$ is passed to the classifier for label prediction:

$$b = \mathrm{LN}(x_I^o). \tag{23}$$

The MSA block is a crucial component, determining the importance of individual patches relative to others in the sequence. MSA computes query ($C$), key ($E$), and value ($Y$) matrices using learned weights $M_{CEY}$. These are used to calculate attention scores:

$$[C, E, Y] = x M_{CEY}, \quad M_{CEY} \in \mathbb{R}^{k \times 3D_E} \tag{24}$$

$$A = \mathrm{softmax}\left(\frac{CE^T}{\sqrt{D_E}}\right), \quad A \in \mathbb{R}^{m \times m} \tag{25}$$

$$\mathrm{SA(x)} = \mathrm{A} \cdot \mathrm{Y}. \tag{26}$$

The MSA block concatenates outputs from all attention heads and passes them through a feed-forward layer with learnable weights:

$$\mathrm{MSA}(x) = \mathrm{Concat}(\mathrm{SA}_1(x); \mathrm{SA}_2(x); \ldots; \mathrm{SA}_t(x)) W, \quad W \in \mathbb{R}^{tD_E \times D}. \tag{27}$$

In this work, a tiny16 vision transformer is utilized. The last three layers (indexing, fully connected, and softmax) are replaced with global average pooling, a new fully connected layer, and a new softmax layer for transfer learning. The modified tiny16 model has three heads and $N \times 192$ hidden dimensions. The architecture is visually presented in Fig. 3.

## Hyperparameter tuning using grey wolf optimization

The grey wolf optimizer (GWO) (*Bazi et al., 2021*) was inspired by the social behaviour of grey wolves, the leadership hierarchy, and pursuit of property in the group. Within their natural habitat, grey wolves typically inhabit groups. The range of group sizes is from five to twelve. A strict social dominance hierarchy is maintained. The most prominent male or female wolves are placed at the apex of the hierarchy as alphas. These individuals are primarily tasked with making decisions regarding the wolf pack's habitat, foraging, resting, and feeding of the wolf pack. Every other pack follows the dominant canines. Beta wolves are the subsequent level of the alpha pack; they execute the directives and exercise dominion over the lower-level wolves. Delta wolves assist alpha and beta wolves in pursuing and investigating prey. They patrol the boundaries of the territory, communicate potential threats to other wolves, and take care of the needs of wounded or vulnerable individuals. Omegas are the lowest category of wolves, which exist outside the hierarchy of other canines. Their social hierarchy predominantly determines wolves' foraging effectiveness. The mathematical modelling of grey wolf social behaviour involves identifying the optimal solution for the prey location and representing the wolf's position

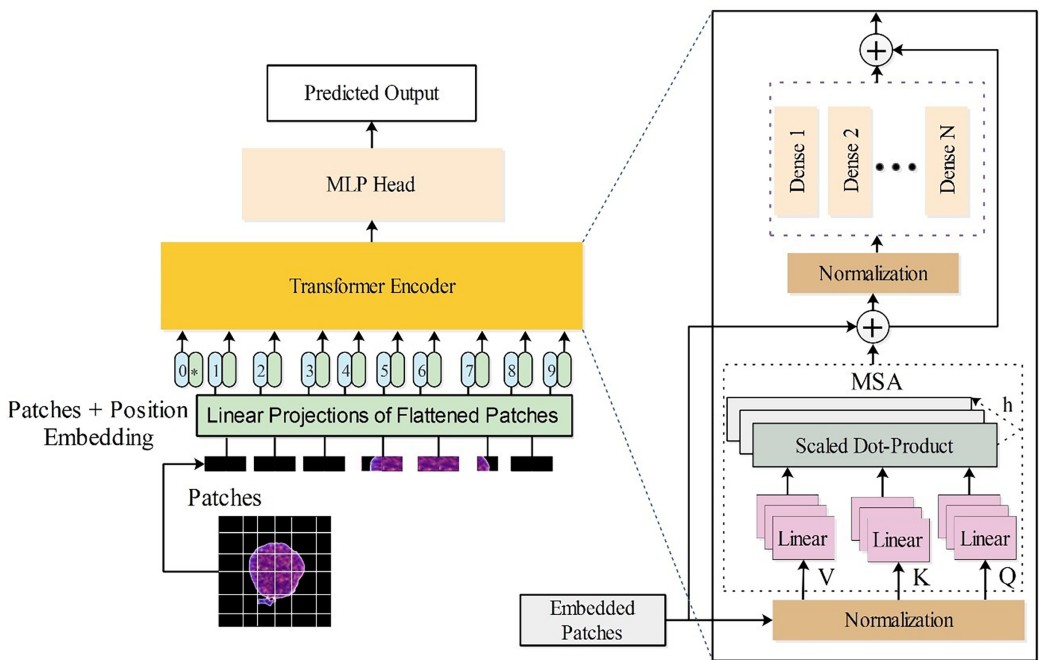

**Figure 3 Customized vision transformer architecture for the classification of leukemia disease.**

in the search space as the solution. Alpha wolves are the most optimal solution, given their proximity to the prey. According to their social hierarchies, the beta and delta wolves represent the next best solution. In space, omega wolves adjust their position during the search process based on the whereabouts of alpha, beta, and delta wolves. Suppose that the position in the space of alpha, beta, delta, and omega is denoted by $\omega_\alpha$, $\omega_\beta$, $\omega_\delta$, and $\omega_\gamma$. Prey surrounding, hunting, attacking, and seeking are essential steps in the GWO. The surrounding is a process by which wolves encircle the prey during hunting time, which is a mathematically formulated equation:

$$\Phi_D = \left| \mathbf{C_1} \cdot \omega_\rho(t) - \omega(t) \right| \tag{28}$$

$$\omega(t+1) = \omega_\rho(t) - \mathbf{C_2} \cdot \Phi_D \tag{29}$$

$$\mathbf{C_1} = 2 \cdot \tau \cdot \mathbf{R_1} - \tau \tag{30}$$

$$\mathbf{C_2} = 2 \cdot \mathbf{R_2} \tag{31}$$

where $\mathbf{C_1}$ and $\mathbf{C_2}$ are the two coefficient vectors, $\omega(t)$ and $\omega_\rho(t)$ represent the position vector of wolves and prey at the current iteration. $\mathbf{R_1}$ and $\mathbf{R_2}$ are two random matrices with the range of $[0, 1]$, and $\tau$ is a matrix whose values decrease over the iteration from 2 to 0. Alpha wolves perform the role of guides during the prey stalking process. Delta and beta wolves are also implicated in this process. It is postulated that these three canines possess knowledge regarding the probable location where sustenance could be encountered. This information facilitates the determination of the three most efficient search agents, which

are subsequently utilized to update the coordinates of other wolves, as represented by equations:

$$\Phi_\alpha = |\mathbf{C_1} \cdot \omega_\alpha - \omega| \tag{32}$$

$$\Phi_\beta = |\mathbf{C_1} \cdot \omega_\beta - \omega| \tag{33}$$

$$\Phi_\delta = |\mathbf{C_1} \cdot \omega_\delta - \omega| \tag{34}$$

$$\omega_1 = \omega_\alpha - \mathbf{T_1} \cdot \Phi_\alpha \tag{35}$$

$$\omega_2 = \omega_\beta - \mathbf{T_2} \cdot \Phi_\beta \tag{36}$$

$$\omega_3 = \omega_\delta - \mathbf{T_3} \cdot \Phi_\delta \tag{37}$$

$$\omega(t+1) = \frac{\omega_1 + \omega_2 + \omega_3}{3}. \tag{38}$$

The attacking phase is equivalent to exploitation and is implemented by the factor $\tau$. When the prey ceases to move, the predatory wolves launch an assault on the defenseless prey. The value of $\mathbf{T}$ is a randomly selected value within the range $[2r, 2r]$, where $r$ is within the range $[-1, 1]$. The process of seeking or exploring the most optimal solution is related to the rising behavior of wolves. Wolves converge after locating prey after diverging in pursuing it. If $|T| > 1$, wolves diverge in search of superior prey; otherwise, if $|T| < 1$, they converge towards the prey. A random $C$ is used to prevent local optima and promote exploration. The method produces random values at the beginning and end stages, promoting impartial investigation. In this research, the proposed models' hyperparameters are considered optimization problems. The objective is to determine the optimal hyperparameters of models using the GWO algorithm. The architecture of the proposed model is mathematically described as:

$$\phi_{\mathrm{acc}} = \mathrm{Network}\left(\psi_{\mathrm{Hp}}, \boldsymbol{\omega}_{\mathrm{weights}}, \Psi_{\mathrm{TD}}\right) \tag{39}$$

$$\rho_{\mathrm{Acc}}^{\mathrm{max}} = \mathrm{Network}\left(\psi_{\mathrm{Hp}}, \boldsymbol{\omega}_{\mathrm{weights}}, \Psi_{\mathrm{TD}}\right), \quad \kappa < \kappa^{\mathrm{max}} \tag{40}$$

where $\psi_{\mathrm{Hp}}$ denotes the list of hyperparameters, $\boldsymbol{\omega}_{\mathrm{weights}}$ represents the weights of the proposed network, and $\Psi_{\mathrm{TD}}$ denotes the training dataset. The objective function is to maximize the accuracy of the network for hyperparameters, as mathematically formulated in Eq. (40). In this work, we employed the grey wolf optimization to tune the proposed models during the training process. the ranges of hyperparameters are described in Table 1.

In this work, we also trained all selected models on static hyperparameters, and the static hyperparameters are learning rate, mini-batch size, epochs, and optimizer having values of 0.001, 16, 200, and SGDM. The proposed PIDSAN4 and tiny ViT models have significantly improved accuracy when the hyperparameters are tuned. The proposed PIDSAN2 improved with 1.9% accuracy, and the tiny ViT improved by 3.00% accuracy, as shown in Fig. 4. The GWO was selected for the hyperparameters tuning due to its balance among the exploration and exploitation, computational efficiency, which aligned well with the lightweight nature of our models. While, other alogrithms such as particle swarm

| Table 1 Hyperparameters for selected optimization. | |
|---|---|
| **Hyperparameters** | **Ranges** |
| Learning rate | [0.00021, 1] |
| Section depth | [0.001, 2.914] |
| L2 regularization | $[1e^{-7}, 1e^{-1}]$ |
| Activation type | ReLU, Sigmoid, Clipped ReLU |
| Dropout | [0.24, 0.91] |

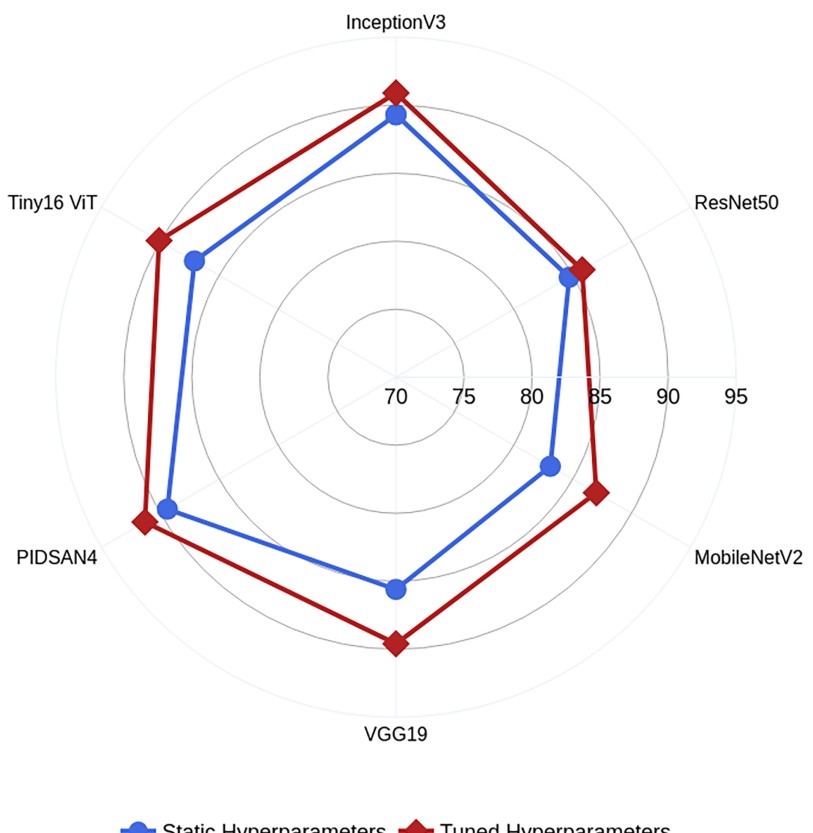

**Figure 4 Comparative analysis among the static and optimized hyperparameters.**

optimization (PSO), firefly optimization, tree growth optimization (TGO) has also outperformed but GWO has consistently showed competitive performance in hyperparameters tuning in recent studies. the process of GWO for tuning are shown in Fig. 5.

## Training process

The selected dataset is divided into multiple ratios: 50:40:10, 60:30:10, and 70:20:10. The 50%, 60%, and 70% of the dataset are utilized for the training process in various experiments, while 40%, 30%, and 20% of data are employed for testing purposes. A total

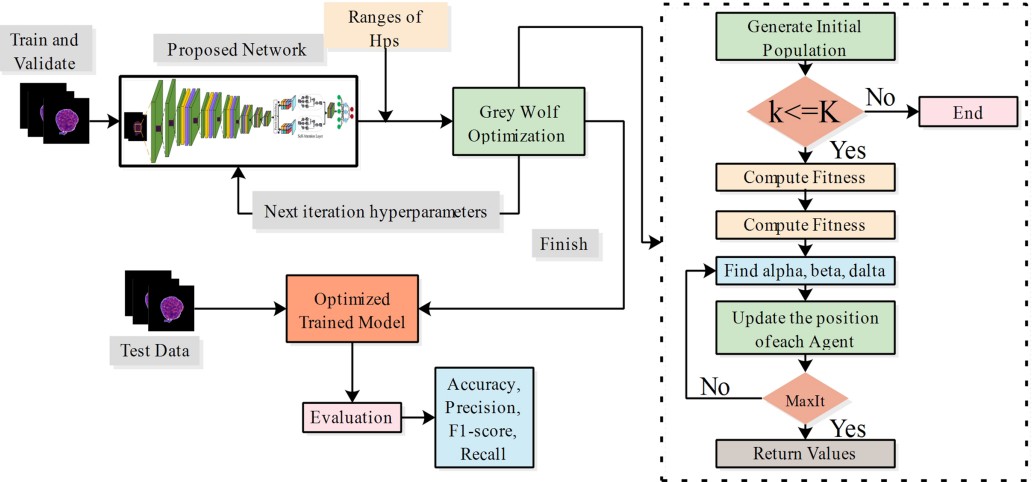

**Figure 5 Process of hyperparameter tuning of proposed models using grey wolf optimization.**

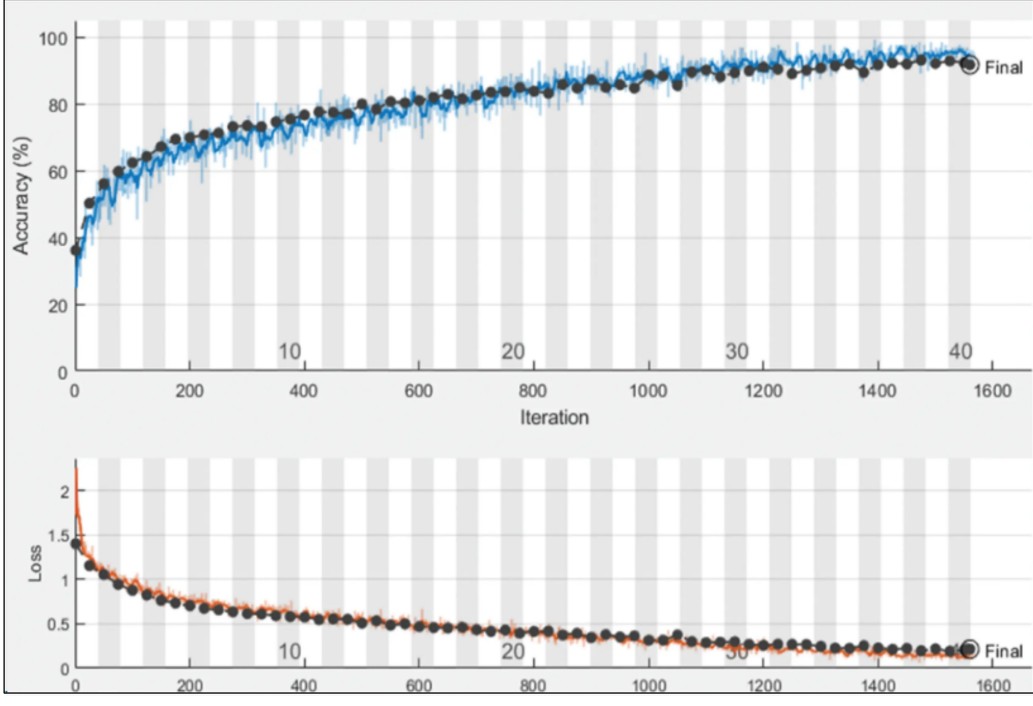

**Figure 6 Training and validation progress on proposed PIDSAN4 model.**

of 10% of the data is used for validation. After analyzing the dataset, it is observed that the selected dataset has an imbalance problem, To address this issue and attain better generalization, we implemented a class weighting method during the training process. Weights are computed inversely proportional to the class occurrences, ensuring that the

model gives more consideration to the underrepresented class. The weights are calculated as:

$$C_\omega = \frac{T_s}{C_N \times C_s} \tag{41}$$

where $C_N$ is the number of classes, $T_s$ is the total count of samples, $C_s$ represents the class samples, and $C_\omega$ is the class weight. The calculated weights are then passed into the loss function to reduce the bias towards the majority class during the learning process.

After designing the proposed models, the proposed models are trained using the training data, as shown in Fig. 6. The hyperparameters for training are selected based on the grey wolf optimization. After training, the trained models are employed for the testing phase.

## EXPERIMENTAL SETUP

The result of the proposed framework is presented in this section. The selected dataset is divided into a 70:20:10 ratio, where 70% of the data is utilized for training, 20% for testing, and 10% for validation. The entire experimental process is carried out using 10-fold cross-validation. The training of the proposed models is dynamically selected using grey wolf optimization. The proposed models were trained using hyperparameters such as learning rate (0.00122), section depth (2), activation type (ReLU), and L2 regularization factor ($1.0003 \times 10^{-9}$). The evaluation parameters are precision, sensitivity, specificity, F-measure, and G-mean. All experiments are performed on MATLAB R2023b, executed on a desktop system configured with an Intel Core i5 processor, 32 GB RAM, 1TB HDD, and an NVIDIA GTX 3060 graphics card.

## EXPERIMENTAL RESULTS

The InceptionV2, ResNet50, MobileNetV2, and VGG19 models were also trained using grey wolf optimization and compared with the proposed models (modified ViT and PIDSAN4). The results of all the trained models are presented in Table 2. From this table, it is observed that the proposed PIDSAN4 model achieved the highest accuracy, which is 0.913. The other parameters are sensitivity (0.892), precision (0.925), F-measure (0.883), and G-mean (0.894), with corresponding values of 0.901. This can be verified by the confusion matrix presented in Fig. 7. This figure illustrates that the PIDSAN4 network confirms strong performance in its claim of generalization. The model achieves good percentage prediction for the Leukemia Blast and Normal Cell samples (92.84% and 91.29%, respectively), demonstrating how the network learned to distinguish these two classes well, which shows its potential for application in further medical diagnosis. Some classification tasks entail erroneous results, such as 104 Leukemia Blast samples misclassified as Normal Cells (7.16%) and 59 Normal Cells misclassified as Leukemia Blast samples (8.71%). The balanced performance in both cases indicates the generalization ability of the model. The second-highest accuracy is achieved by the modified ViT from the listed models, with an accuracy of 0.901, sensitivity of 0.897, specificity of 0.897, precision of 0.889, F-measure of 0.881, and G-mean of 0.87.

**Table 2 Comparison of pre-trained models and proposed models on the leukemia cancer dataset.**

| Models | Accuracy | Sensitivity | Specificity | Precision | F-measure | G-mean |
|---|---|---|---|---|---|---|
| InceptionV3 | 0.909 | 0.904 | 0.901 | 0.895 | 0.894 | 0.893 |
| ResNet50 | 0.858 | 0.841 | 0.844 | 0.852 | 0.853 | 0.851 |
| MobileNetV2 | 0.874 | 0.872 | 0.861 | 0.877 | 0.864 | 0.871 |
| VGG19 | 0.896 | 0.887 | 0.893 | 0.884 | 0.871 | 0.894 |
| Modified ViT | 0.901 | 0.891 | 0.897 | 0.889 | 0.881 | 0.871 |
| Proposed PIDSAN4 | 0.913 | 0.892 | 0.925 | 0.883 | 0.894 | 0.901 |

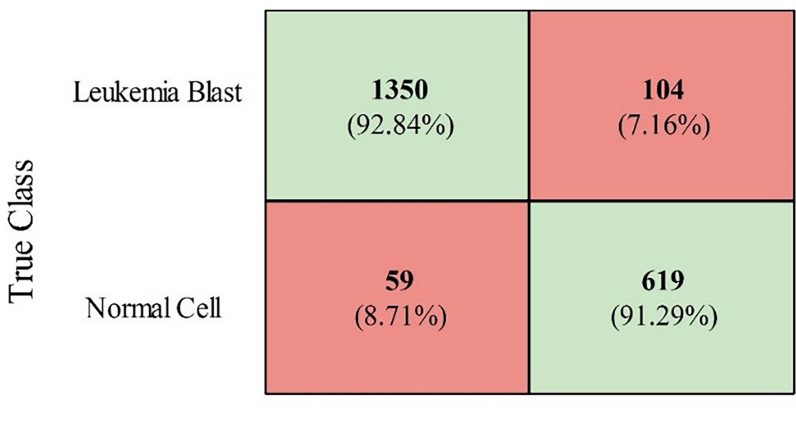

**Figure 7 Confusion matrix of proposed PIDSAN4 on the selected dataset.**

For further clarification, the standard error of the mean (SEM) and confidence interval (CI) were measured using the precision of the models. In this method, the confidence interval is 95%, where the z-score is 1.96, and the test sample size is 2,131. The SEM and CI are calculated for each model, as described in Table 3. The SEM and CI are calculated using Eqs. (43) and (44).

$$M_{SEM} = \frac{\sigma}{\sqrt{N}} \tag{42}$$

$$M_{SEM} = \sqrt{\frac{M_p(1 - M_p)}{N}} \tag{43}$$

$$M_{CI} = M_p \pm z_{\text{score}} \times M_{SEM} \tag{44}$$

where $\sigma$ is the standard deviation, calculated as $\sqrt{M_p(1 - M_p)}$, where $M_p$ is the precision and $(1 - M_p)$ is the error proportion of each model. Table 3 presents the results of the SEM and CI method. According to the table, the InceptionV3 model has the highest precision (0.895) among the models, and its small confidence interval [0.882–0.908] suggests that it is highly reliable. These results show that InceptionV3's low error rate of

**Table 3 Standard error of the mean and confidence interval for each model.**

| Models | $M_p$ | $(1 - M_p)$ | $M_{SEM}$ | $M_{CI}$ (95%) |
|---|---|---|---|---|
| InceptionV3 | 0.895 | 0.105 | 0.00665 | [0.882–0.908] |
| ResNet50 | 0.852 | 0.148 | 0.00779 | [0.837–0.867] |
| MobileNetV2 | 0.877 | 0.123 | 0.00718 | [0.863–0.891] |
| VGG19 | 0.884 | 0.116 | 0.00696 | [0.871–0.897] |
| Modified ViT | 0.889 | 0.111 | 0.00686 | [0.876–0.902] |
| Proposed PIDSAN4 | 0.883 | 0.117 | 0.00698 | [0.870–0.896] |

0.105 was well considered to predict correct classifications. After this model, the Modified ViT achieved a precision of 0.889, showing that transformer-based architectures can also achieve competitive results with a similarly narrow CI [0.876–0.902]. MobileNetV2 and VGG19 performed well with precision scores of 0.877 and 0.884, indicating their stability with relatively narrow CI intervals. Even though ResNet50 had slightly lower precision (0.852) and a wider margin of error (0.148), it still serves as a useful baseline, especially in setups with constrained computational resources. The proposed PIDSAN4 model also performed well, achieving a precision of 0.883 and demonstrating strong stability with a CI of [0.870–0.896]. However, relatively fitted confidence intervals imply that in most cases, models have successfully attained high precision, making them worthy of real-time applications where reliability is critical.

## Ablation study

After optimizing the hyperparameters, all the selected models are trained using various optimizers with the same configurations, as shown in Table 4. According to this table, it is observed that the proposed models achieved higher accuracy with the optimizer of SGDM and the learning rate of $1.22 \times 10^{-3}$. The training time of all the selected models is also noted, and it is noted that InceptionV3 has the highest training time, which is 13.53 h. The proposed model completed its training within 7.98 h. The second-lowest training time is 8.57 h, obtained by the MobileNetV2 model. Three experiments were performed using three ratios of data: 50:40:10, 60:30:10, and 70:20:10. The 50%, 60%, and 70% of the dataset are utilized for the training process in various experiments, while 40%, 30%, and 20% of data are employed for testing purposes, and 10% of data is used for validation. Figure 8 highlights the relationship between data splits and model performance, raising issues of underfitting, overfitting, and generalization considerations. Models such as Tiny16 ViT and the proposed PIDSAN4 show consistent performance improvements as training data increases, showing better generalization capabilities. Older architectures such as ResNet50 and VGG19 show lower precision and scalability with more training data, suggesting that they may be inadequate due to their limited ability to learn complex patterns from data sets. For the split of 50:40:10, lower training data may result in inadequate adaptation for most models, as shown in the modest accuracy values, except PIDSAN4 and Tiny16 ViT, which still achieve high performance. As the distribution increases to 70:20:10, most models improve accuracy. Still, the distance between PIDSAN4/Tiny16 ViT and other

**Table 4 Selected models are trained on different optimizers with the same configurations.**

| Models | Adam | SGDM | RMSprop | Training time (H) |
|---|---|---|---|---|
| InceptionV3 | 0.8561 | 0.909 | 0.8714 | 13.53 |
| ResNet50 | 0.8277 | 0.858 | 0.8416 | 10.52 |
| MobileNetV2 | 0.8471 | 0.874 | 0.8661 | 8.57 |
| VGG19 | 0.8631 | 0.896 | 0.8514 | 11.17 |
| Modified ViT | 0.8741 | 0.901 | 0.884 | 12.41 |
| Proposed PIDSAN4 | 0.8319 | 0.913 | 0.8462 | 7.97 |

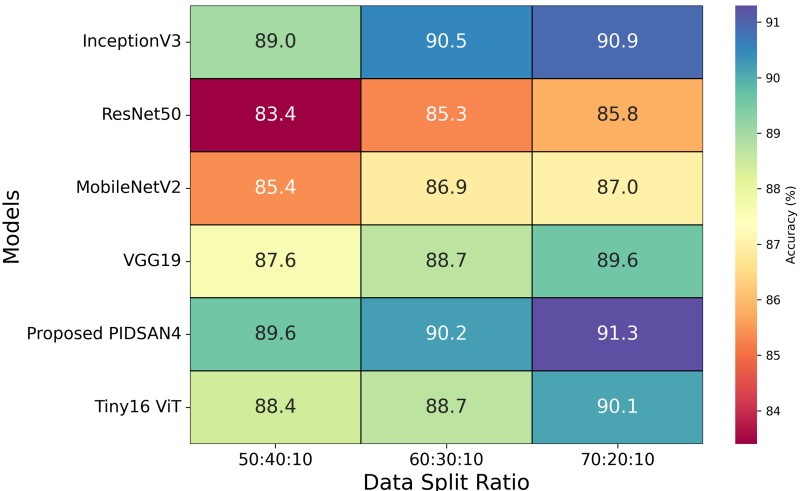

**Figure 8 Results of selected models based on different dataset split ratios.**

models increases, indicating that older models may be too suitable or unable to generalize due to their weak adaptability to larger data sets. The high performance of PIDSAN4 and Tiny16 ViT in all partitions shows that these models effectively balance generalization based on modern architectural advances.

A comprehensive comparison of proposed models and the state-of-the-art models has been conducted in this section. Table 5 illustrates the comparison based on the number of layers, number of parameters, and size of the models. From this table, Inception V3 has 316 total layers with 23.9 million parameters and 23.9 MB in size. The ResNet50 and MobileNetv2 have 177 and 154 layers with 25.6 and 3.5M parameters, respectively. Both models are 96 and 13 MB in size. VGG19 has 47 layers with 143.6M parameters and 535 MB in size. The proposed PIDSAN4 model has 107 layers with 7.5M parameters and 16.4 MB in size. The other proposed model, Tiny ViT, has 143 layers with 5.7M parameters and only 22.6 MB in size. As a result, concerning parameters, the proposed model is much more lightweight than InceptionV3, ResNet50, and VGG19. The receiver operating characteristic (ROC) curve shows all six of the models. InceptionV3, ResNet50, MobileNetV2, VGG19, Modified ViT, and PIDSAN4. The ROC curve plots the Sensitivity against the False Positive Rate for the model. The three models that performed the best in the evaluation and were most effective were InceptionV3, Modified ViT, and PIDSAN4

**Table 5 Comparison of the proposed PIDSAN4 with state-of-the-art models.**

| Models | No. of layers | No. of parameters | Size in MB |
|---|---|---|---|
| InceptionV3 | 316 | 23.9M | 23.9 MB |
| ResNet50 | 177 | 25.6M | 96 MB |
| MobileNetV2 | 154 | 3.5M | 13 MB |
| VGG19 | 47 | 143.6M | 535 MB |
| Modified ViT | 143 | 5.7M | 22.6 MB |
| Proposed PIDSAN4 | 107 | 7.5M | 16.4 MB |

**Table 6 Comparison of ViT architectures with proposed models based on performance metrics.**

| ViTs | Architecture type | Precision | Recall | F1-score | Specificity | Accuracy |
|---|---|---|---|---|---|---|
| ConvNeXt-t | CNN-Transformer Hybrid | 0.874 | 0.871 | 0.872 | 0.889 | 0.894 |
| DaViT-S | Dual attention ViT | 0.866 | 0.869 | 0.867 | 0.877 | 0.882 |
| CrossViT-s | Cross-Attention ViT | 0.871 | 0.873 | 0.872 | 0.882 | 0.888 |
| Proposed PIDSAN4 | CNN + Self-Attention | 0.883 | 0.892 | 0.894 | 0.925 | 0.913 |
| Modified TinyViT | Vision transformer | 0.889 | 0.897 | 0.881 | 0.897 | 0.901 |

with an area under the curve (AUC) of 0.91, indicating that they are excelling at class discrimination. VGG19 was close behind with an AUC of 0.90 indicating it also performed strongly. MobileNetV2 and ResNet50 performed comparatively weaker than their peers with an AUC of 0.87 and 0.86 respectively suggesting that they are slightly less effective than others for the purpose of demonstrating class discrimination ability for this evaluation setting. The ROC is presented in Fig. 9.

Table 6 shows a comprehensive comparison of various ViTs and hybrid models using five relevant classification metrics of precision, recall, F1-score, specificity, and accuracy. The proposed PIDSAN4 model which is based on hybrid architecture that combines CNN with self-attention, showed more balance and superior performance across all metrics. It had the highest F1-score (0.894), which indicates the best precision and recall tradeoff. It is also superior to the other models in specificity (0.925) and accuracy (0.913). This shows its ability to correctly classify both the positive and negative samples, which is especially salient for maintaining low false positive rates in a high consequence classification task. The modified TinyViT made no changes to a pure ViT architecture, showed the highest precision (0.889) and recall (0.897) which show its strength in correctly identifying true positive samples. However, while it demonstrated significant sensitivity in detection, it shows a somewhat weaker F1-score (0.881) and accuracy (0.901) compared to that of PIDSAN4 which indicates a slight imbalance in false positive and false negative handling. While a slight imbalance should not affect its reliability, it may have relevance in some uses where consistent performance across classes is pertinent. The ConvNeXt-t and CrossViT-s models are CNN-Transformer hybrid approaches demonstrated F1-scores nearly identical at 0.872. While they both showed reasonable performance comparably, both also underperform compared to PIDSAN4 in specificity and accuracy. DaViT-S, which showcased dual attention mechanisms, showed the lowest specificity (0.877) and accuracy

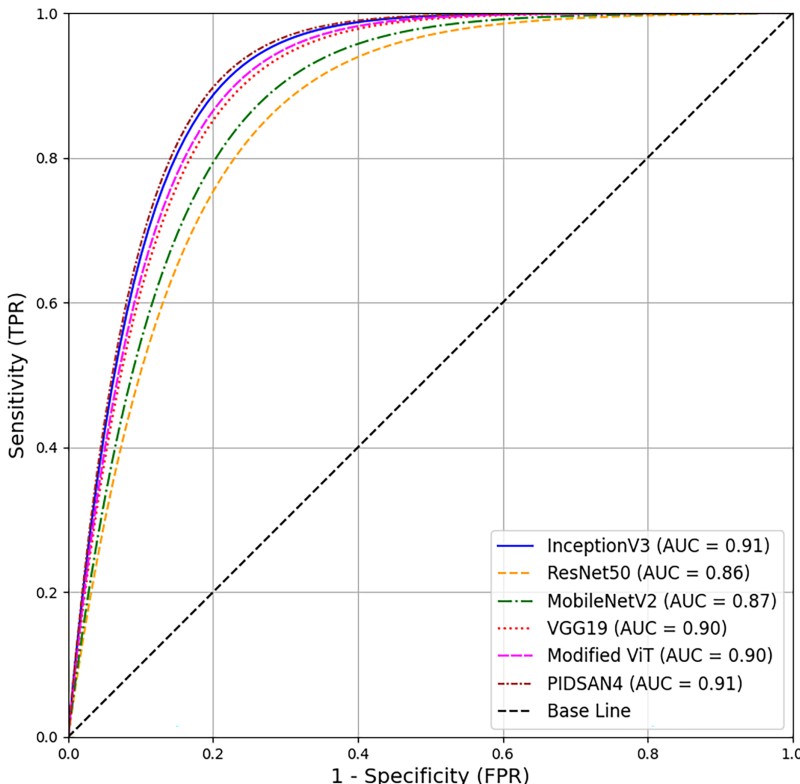

**Figure 9 ROC based comparison of proposed models with state-of-the-art methods.**

(0.882) of any of the models, therefore limiting its ability to correctly identify negatives or reject potential negative instances. Larger proportion of false negatives may induce higher levels of false alarms in potential real-world applications.

Table 7 describes a useful comparison of a number of metaheuristic optimization methods for hyperparameter tuning and their effect on the performance of the proposed PIDSAN4 model. The results show that a properly tuned hyperparameter tuning approach utilizing metaheuristics can provide some performance benefits regarding classification performance. Compared to the without tuned performance of the model accuracy is 0.881, F1-score is 0.891, the correctly tuned options optimize training dynamics that allow the classifiers to refine predictive accuracy and generalization. The GWO method is by far the best evaluated optimizer with respect to the metrics that are used. It have both the highest F1-score (0.894) and highest accuracy (0.913). The GWO must have performed accordingly because it is able to search the search space, and find optimal combinations of hyperparameters to use while training the networks in a way that improved their learning. The chimp optimizer provided strong accuracy which is 0.879, F1-score is 0.868 performance, followed closely by firefly optimizer accuracy (0.881). While the TGO ACO are effective in terms of performance improvements, they did not provide as much performance benefit as GWO and firefly, which indicates GWO and Firefly have better search features compared to the ACO and TGO. TGO performed worse than the baseline.

**Table 7 Performance comparison of optimization algorithms with and without hyperparameter tuning.**

| Optimizations | With tuning | Precision | Recall | F1-score | Accuracy |
|---|---|---|---|---|---|
| W/o tuning | – | 0.879 | 0.887 | 0.891 | 0.881 |
| Tree growth (TGO) | ✓ | 0.867 | 0.859 | 0.863 | 0.874 |
| Firefly optimization | ✓ | 0.876 | 0.862 | 0.869 | 0.881 |
| Ant colony (ACO) | ✓ | 0.869 | 0.858 | 0.863 | 0.875 |
| Chimp optimization | ✓ | 0.872 | 0.865 | 0.868 | 0.879 |
| Grey wolf (GWO) | ✓ | 0.883 | 0.892 | 0.894 | 0.913 |

Figure 10 demonstrates the comparative performance of the PIDSAN4 model and the Modified ViT model while they both operated under varying intensities of image noise with the purpose of examining the performance of each model across three levels of Gaussian noise (1%, 3% and 5%). The performance results demonstrated that as the noise intensity increased the overall performance for each model would gradually decrease; however, the PIDSAN4 model maintained a better performance under all levels of noise. At noise level 1%, the PIDSAN4 model demonstrated the highest performance with an accuracy of 90.8% and an F1-score of 88.9%. Comparatively, the modified ViT recorded an accuracy of 89.4% and F1-score of 87.1% for the model performance. When the noise intensity level is raised to a level of 3%, the PIDSAN4 model again maintained a better performance with a model accuracy of 89.2% and an F1-score of 87.4% which compared to the modified ViT's accuracy of 87.9% and F1-score of 85.3% respectively. By noise level 5% the PIDSAN4 model still have relatively better performance with an accuracy of 87.1% and F1-score of 86.6% while the modified ViT have an accuracy of 85.4% and F1-score of 83.3%.

## Discussion

The results clearly show that the proposed PIDSAN4 and modified Tiny ViT models are more efficient on leukemia cancer datasets than state-of-the-art architectures such as InceptionV3, ResNet50, MobileNetV2, and VGG19. In particular, the PIDSAN4 model achieved the highest precision of 91.3% and a balanced sensitivity (89.2%), specificity (92.5%), and precision (83.8%). Its superior measurement demonstrates its generalization capability, as evidenced by the confusion matrix, in which it accurately classified 92.84% of leukemia blast samples and 91.29% of normal cells. These results demonstrate the network's ability to effectively learn complex patterns, minimize classification errors, and maintain balance between both classes.

The modified Tiny ViT achieves slightly reduced accuracy 0.901 compared to InceptionV3 0.909, however, its inclusion in our investigation was as part of an intentional tradeoff between performance, computation, and interpretability in model building. The Tiny ViT is much smaller which has 5.7M parameters compared to InceptionV3 which is 23.9M and requires less time to train 12.41 *vs.* 13.53 h. This compactness means it is more appropriate for real-time use in less resource-rich environments, such as portable diagnostic devices and clinics. A transformer model enhances explainability since attention

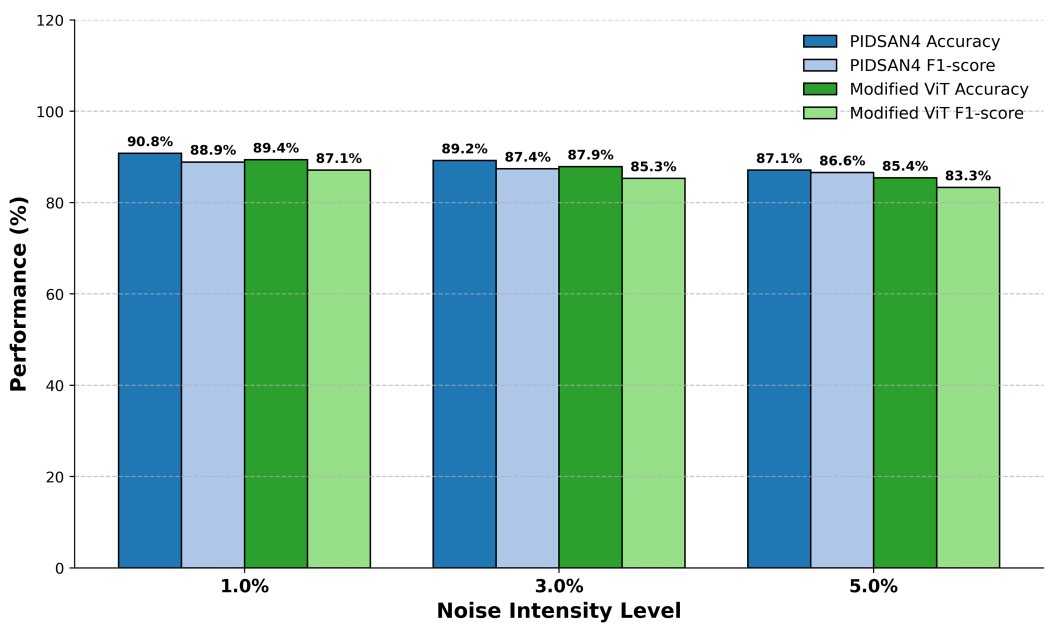

**Figure 10 Evaluation of proposed models on noisy data.**

has been shown to help understand decision-making in some cases and we used LIME visualization to understand the decisions made by the model as shown in Fig. 8. Our framework contributes to model development, including Tiny ViT for model evaluation, and the view is not that Tiny ViT replaces a standard CNN, such as the InceptionV3 model, but can contribute to a dual-model strategy whereby models based on different architecture types can enable even more features and provide a strong basis for exploration in clinical applications.

Using modern architectural advances in AI models, such as including lightweight layers and efficient feature extraction mechanisms, the PIDSAN4 network has overcome heavy architectures such as VGG19, which suffers from overcomplicated fitting due to its large parameter sizes and limited scale. PIDSAN4 and Tiny ViT models offer superior accuracy and computational efficiency, making them suitable for integration into medical diagnostic workflows. Their lightweight architecture can be deployed in resource-limited environments, such as portable diagnostics and hospitals with limited computational resources. Balanced classification performance minimizes critical errors such as false negatives (false classification of leukemia cells as normal), reducing the risk of missed diagnosis. This high reliability and adaptability can improve early leukemia detection, support clinical decisions, and personalized treatment planning.

Figure 11 depicts LIME visualizations for five randomly chosen samples processed by the proposed Tiny Vision Transformer (Tiny16), and the PIDSAN4 model. The process of LIME consists of perturbing the input image and measuring how the prediction changes. It then fits a simple interpretable model locally around each sample to model the contribution of relevant image regions to a final decision. In these visualizations,

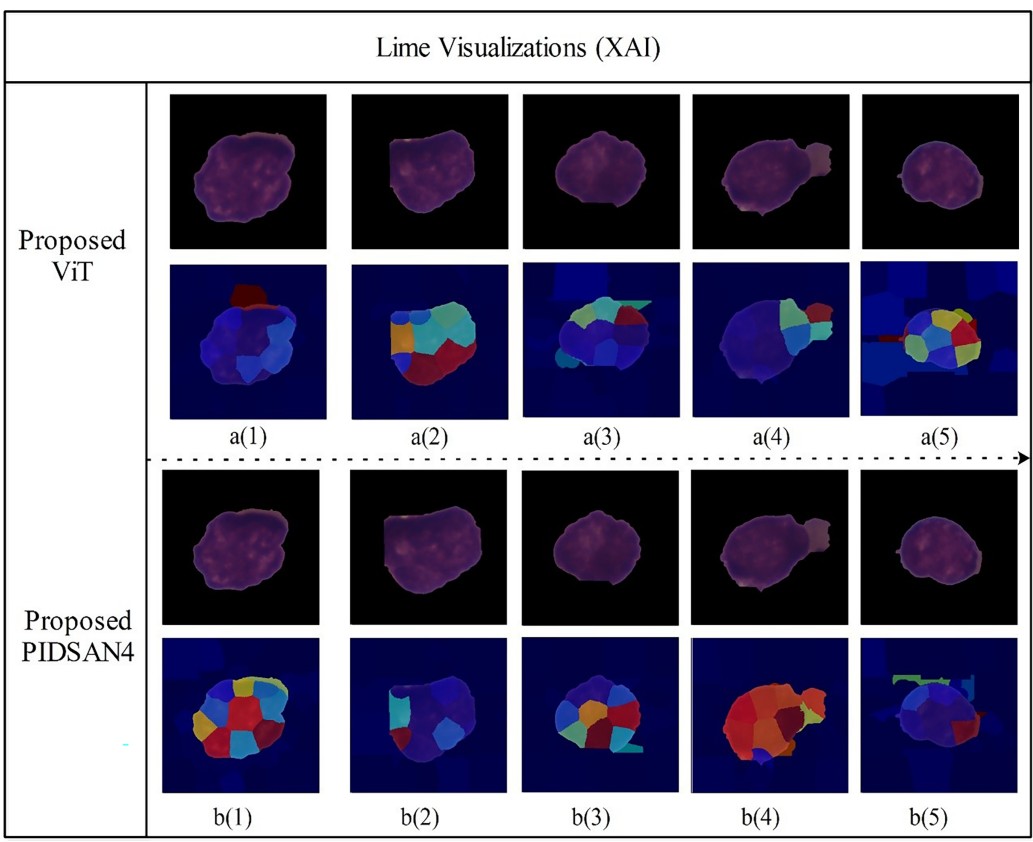

Figure 11 Visual explanation results of proposed models using Lime XAI.

superpixels such as small colored areas are those areas of the image that LIME estimated contributed most to a prediction decision made by the model. The warmer colors such as red, yellow, orange indicate the image regions that highly support a predicted class and the cooler colors like blue, green are areas of less and even negative influence. From the top row, the Tiny ViT model appeared to distribute attention more widely across the image area and focus attention on several, dispersed areas. This could be related to the Tiny ViT ability to apply long-range dependencies, and to attend to a global context and certainly less local attention than the PIDSAN4 model was able to apply. The PIDSAN4 model showed more local and focused attention across the same areas in the image. The LIME maps of the PIDSAN4 model showed that the regions that consistently highlighted the core nuclear regions and showed the red and yellow regions of attention. This consistent application of focus by the PIDSAN4 model reflects the model's reliance on these specific morphological features of leukemic cells such as enlarged nuclei, increased chromatin density.

## COMPARISON WITH SOTA METHODS

The comparison table focuses on the comprehensive comparison of the current state-of-the-art methodology (SOTA) and the proposed models, as shown in Table 8. In existing

**Table 8 Comprehensive comparison among the proposed and SOTA techniques.**

| Ref | Year | Methodology | Hp's tuning | Accuracy |
|---|---|---|---|---|
| *Park et al. (2024)* | 2024 | EfficientNet-B2 | – | 87.7% |
| *Ilyas et al. (2024)* | 2024 | Fisher's linear discriminant analysis | – | 89.6% |
| *Wang et al. (2024)* | 2024 | Survival-SVM and random survival forest | – | 89.2% |
| *Huang & Huang (2024)* | 2024 | Ensemble CNNs | Yes | 91.3% |
| *Wibowo, Rianto & Unjung (2024)* | 2024 | EfficientNetV2M | Yes | 87.0% |
| *Rahmani et al. (2024)* | 2024 | Combined DL models | Yes | 90.7% |
| *Vasumathi et al. (2025)* | 2025 | ResNet50 with DTL | – | 82.69% |
| Proposed ViT | – | ViT | – | 90.1% |
| Proposed PIDSAN4 | – | PIDSAN4 | – | 91.3% |

methods, the (*Huang & Huang, 2024*) ensemble CNN method has achieved the highest accuracy of 91.3%. This success is due to the integrated strategy that combines several CNN models to utilize complementary features and increase classification performance. *Rahmani et al. (2024)* successfully implemented the combined deep learning model (DL), which achieved 90.7% accuracy because the DL architecture was integrated into multiple systems. Methods such as survival SVM and random survival forest of *Wang et al. (2024)* (89.2%) and *Ilyas et al. (2024)* Fisher's linear discriminant analysis (89.6%) demonstrate the limitations of classical machine learning techniques in processing high-dimensional medical imaging data. The proposed PIDSAN4 model is notable in terms of computational efficiency compared to *Huang & Huang (2024)* CNN Ensemble with a precision of 91.3%. Unlike CNN Ensemble, it uses a lightweight design. It incorporates modern architectural advances such as efficient feature extraction layers and hyperparameter adjustment to deliver high accuracy without the overhead of the ensemble method, unlike the CNN Ensemble, which uses its architecture and resources for its architecture.

*Vasumathi et al. (2025)* presented a framework based on ResNet50 using DTL and employed some tradition machine learning approaches such as random forest and SVM for the classification of leukima blast cells. they achieved 82.69% highest accuracy.

The proposed ViT model also shows competitive performance with 90.1% accuracy, highlighting the potential of transformer-based architecture in medical classification tasks. Compared to EfficientNet-B2 from *Park et al. (2024)*, which achieved 87.7%, and *Rahmani et al. (2024)*, EfficientNetV2M gained 87.0% accuracy, the proposed models emphasize the limitations of standard vector methods for capturing complex patterns in medical imaging data. This comparison strengthens the suitability of the proposed PIDSAN4 and ViT models for leukemia classification tasks, combining high precision, computational efficiency, and adaptation to the real medical diagnostic environment.

## CONCLUSION

Leukemia cells generally exhibit irregular shapes, large nuclei and prominent nuclei compared to normal cells, which requires advanced diagnostic techniques. Therefore, the parallel Inverted dual self-attention network (PIDSAN4) was designed to address the

challenges of leukemia cell, such as irregular shapes and large nuclei. The proposed architecture consists of four inverted residual blocks and self-attention mechanisms, enabling efficient extraction of features and learning long-term dependency. This lightweight design only uses 7.5M parameters, ensures computational efficiency, and can be deployed in resource-limited environments and also modified ViT is proposed (ViT-Tiny16) to capture complex and subtle features by attention mechanism. The proposed model demonstrated superior performance, achieving an accuracy of 0.913, sensitivity of 0.892, specificity of 0.925, precision of 0.883, F-measure of 0.894, and G-mean of 0.901. Comparisons with state-of-the-art pre-trained models and ViTs revealed that the proposed model led to improved diagnostic accuracy and prominence of the potential of the proposed automated technique to assist medical experts in achieving higher diagnostic precision and efficiency in the detection of leukemia cancer and lime is employed to further interpret the models to insure the decisions. The framework is evaluated with limited datasets, which may not adequately represent the diversity of microscope images in clinical practice.

Future research will focus on validating models in larger, more diverse datasets to ensure generalization. Efforts to optimize the computational efficiency of architectures would facilitate the wider adoption. The integration of advanced explanation AI techniques beyond LIME could further improve transparency. A study of the inclusion of multimodal data such as clinical data or genetic information can also extend the applicability and diagnostic accuracy of the framework.

### Funding
This research was funded by Princess Nourah bint Abdulrahman University Researchers Supporting Project number (PNURSP2025R435), Princess Nourah bint Abdulrahman University, Riyadh, Saudi Arabia. The funders had no role in study design, data collection and analysis, decision to publish, or preparation of the manuscript.

### Grant Disclosures
The following grant information was disclosed by the authors:
Princess Nourah bint Abdulrahman University Researchers Supporting, Riyadh, Saudi Arabia: PNURSP2025R435.

### Competing Interests
The authors declare that they have no competing interests.

### Author Contributions
- Shams ur Rehman conceived and designed the experiments, performed the experiments, prepared figures and/or tables, and approved the final draft.
- Robertas Damaševicius conceived and designed the experiments, performed the experiments, performed the computation work, prepared figures and/or tables, authored or reviewed drafts of the article, and approved the final draft.

- Hassan Al Sukhni analyzed the data, prepared figures and/or tables, and approved the final draft.
- Abeer Aljohani analyzed the data, authored or reviewed drafts of the article, and approved the final draft.
- Ameer Hamza conceived and designed the experiments, performed the experiments, performed the computation work, prepared figures and/or tables, and approved the final draft.
- Deema Mohammed Alsekait analyzed the data, prepared figures and/or tables, authored or reviewed drafts of the article, and approved the final draft.
- Diaa Salama AbdElminaam performed the computation work, authored or reviewed drafts of the article, and approved the final draft.

## Data Availability

The dataset is available at Zenodo: Hamza, A. (2025). Leukemia Classification [Data set]. Zenodo. https://doi.org/10.5281/zenodo.15489534.

The code is available at GitHub and Zenodo:

- https://github.com/Ameer-Hamxa105/LeukimaCode.git.

- Hamza, A. (2025). Leukemia Classification Using Deep Learning and Grey Wolf Optimization. https://doi.org/10.5281/zenodo.15476028.

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
