# Peer review of "A novel deep learning based approach with hyperparameter selection using grey wolf optimization for leukemia classification and hematologic malignancy detection"

_PeerJ Computer Science, doi:10.7717/peerj-cs.3160_

## Round 0.1 · original submission · Major Revisions

Please address all of the reviewers' points.

·

Basic reporting

-

Experimental design

1. There is no preprocessing involved, which is very much required for accurate results. The size of the images is not focused.
2. The evaluation method is done by using usual CNN models and Grey Wolf optimization. Many research works are already available in the same methodology. Authors should highlight the novelty of their work.

Validity of the findings

1. Novelty is missing.
2. The modified Tiny ViT performance is less than InceptionV3. Being a medical image processing work, accuracy is more important than the complexity of the network.
3. There is no specification of colors used in Figure 1.
4. So many optimization algorithms have evolved nowadays than a grey wolf. Why didn't the authors compare the Grey Wolf algorithm with others?

Additional comments

1. The input image split ratio is given as a table and also as a figure. Authors can avoid repetition of information.

2. There is no clarity of explanation in Figure 8.

3. All the results are obtained nicely. But there is no inference of the results throughout the paper.

4. The Grey Wolf algorithm is most common. There is no need for an explanation of the step-by-step procedure.

5. Authors can use some interesting graphs, like Glyph, spider plots, etc., to show their results instead of the usual bar chart.

Reviewer 2 ·

Basic reporting

The Introduction should be revised and highlight the research gap.
Highlight the role of deep learning and ViT-based models in the the manuscript to underline the technological significance of the work “Enhancing Blood Cell Diagnosis Using Hybrid Residual and Dual Block Transformer Network”, “Blood Cancer Detection Leveraging Deep Learning Fusion Techniques: An Optimized Model Based on VGG16 and AlexNet”, “Morphological diagnosis of hematologic malignancy using feature fusion-based deep convolutional neural network”, “Feature Fusion based Deep Learning method for Leukemia cell classification”

Experimental design

The CNN model description should be revised.
Please provide the complexity of the model.
How proposed model react to the noises
Define each variable value used in the study.
How Gray Wolf Optimization improved results. An ablation study can be added.
Visualize the learning process of the model, including training and validation curves, and analyze any observed trends to provide deeper insights into the optimization behavior.
The results would benefit from a more rigorous statistical analysis to support the claims of performance improvements. Including confidence intervals or significance testing would lend credibility to the findings.

Validity of the findings

Compared the results of the proposed model with deep learning and ViT-based methods like ConvNeXt, DaViT and CrossViT.
The ROC Plot-based comparison of the methods discussed in Table 6 can be added.
The method should be validated on another dataset for the generalization of the model.

Additional comments

The conclusion should be revised.
How many heads did you select in the proposed study?
Please specify the feature dimension and patch size used in the transformer.
The discussion section should be improved.
Lime XAI need further explanation.

Reviewer 3 ·

Basic reporting

. Highlight your contributions more precisely.
2. Add one paragraph to discuss a more recent non-deep learning-based work, including the following work, and discuss its limitations in the introduction section to present the importance of deep learning methods.

Experimental design

-

Validity of the findings

Compare the proposed method with existing at least deep learning-based methods published in 2024 or 2025 (at least compare with the presented performance in these works with giving proper citation).

Additional comments

Highlight the future scopes in the conclusion.

---

## Round 0.2 · Minor Revisions

Dear Authors,
Your paper has been revised. It needs minor revisions before being accepted for publication in PEERJ Computer Science. More precisely;

1) The considered dataset shows a data imbalance between normal and malignancy images. In their analysis, the authors should consider an equal amount of images for both.

2) The ROC curve slope shown in Figure 9 has a strange behavior. The authors should check it.

·

Basic reporting

It is well and good in the revised manuscript.

Experimental design

The reviewer's comments are brilliantly incorporated in the revised manuscript.

The dataset is collected from 118 patients, and it has two classes, including normal cells and leukemia blasts. The normal class contains 3389 microscopic images, and the leukemia blast consists of 7272 microscopic images. It shows data imbalance between normal and malignancy images. The authors should consider an equal amount of images for both.

Validity of the findings

The ROC curve slope shown in Figure 9 is abnormal. AUC is less in the given figure 9. It should come in a reverse manner (refer to attachment)

Reviewer 2 ·

Basic reporting

The manuscript has been improved.

Experimental design

The experimental section has been improved.

Validity of the findings

The validity and findings have been improved.

Additional comments

The ROC curve should be above the baseline. Please check

---

## Round 0.3 · accepted · Accept

Dear Authors,
Your paper has been revised. It has been accepted for publication in PEERJ Computer Science. Thank you for your fine contribution.

·

Basic reporting

Good

Experimental design

Good

Validity of the findings

Good

Additional comments

All the reviewer comments are considered and excuted

Reviewer 3 ·

Basic reporting

1. Highlight your contributions more precisely.
2. Add one paragraph to discuss a more recent non-deep learning-based works including the following work and discuss their limitations in the introduction section to present the importance of the deep learning methods.
3. Add one paragraph to discuss more recent advancements in deep learning for other computer vision applications, specifically for medical applications

Experimental design

no comment

Validity of the findings

no comment

Additional comments

Highlight the limitations of existing works discussed in the literature (related work).